# Fluorescence, Absorption, Chromatography and Structural Transformation of Chelerythrine and Ethoxychelerythrine in Protic Solvents: A Comparative Study

**DOI:** 10.3390/molecules27154693

**Published:** 2022-07-22

**Authors:** Jinjin Cao, Yanhui Zheng, Ting Liu, Jiamiao Liu, Jinze Liu, Jing Wang, Qirui Sun, Wenhong Li, Yongju Wei

**Affiliations:** 1Department of Environmental and Chemical Engineering, Hebei College of Industry and Technology, Shijiazhuang 050091, China; caojj1219@foxmail.com (J.C.); liuting.0996@foxmail.com (T.L.); 18633803153@sohu.com (J.L.); 18032851441@foxmail.com (J.L.); 2Department of Preschool and Arts Education, Shijiazhuang Vocational College of Finance & Economics, Shijiazhuang 050061, China; 3College of Chemistry and Material Science, Hebei Normal University, Shijiazhuang 050024, China; 15303230376@sohu.com (J.W.); 13931183175@foxmail.com (Q.S.); weiyju@126.com (Y.W.)

**Keywords:** chelerythrine, ethoxychelerythrine, fluorescence, absorbance, structural transformation

## Abstract

Chelerythrine (CH) and ethoxychelerythrine (ECH) are chemical reference substances for quality control of Chinese herbal medicines, and ECH is the dihydrogen derivative of CH. In this study, their fluorescence and absorption spectra, as well as their structural changes in different protic solvents were compared. It was observed that their emission fluorescence spectra in methanol were almost the same (both emitted at 400 nm), which may be attributed to the nucleophilic and exchange reactions of CH and ECH with methanol molecules with the common product of 6-methoxy-5,6-dihydrochelerythrine (MCH). When diluted with water, MCH was converted into CH, which mainly existed in the form of positively charged CH^+^ under acidic and near-neutral conditions with the fluorescence emission at 550 nm. With the increase of pH value of the aqueous solution, CH^+^ converted to 6-hydroxy-5,6-dihydrochelerythrine (CHOH) with the fluorescence emission at 410 nm. The fluorescence quantum yields of MCH and CHOH were 0.13 and 0.15, respectively, and both the fluorescence intensities were much stronger than that of CH^+^. It is concluded that CH and ECH can substitute each other in the same protic solvent, which was further verified by high-performance liquid chromatography. This study will help in the investigation of structural changes of benzophenanthridine alkaloids and will provide the possibility for the mutual substitution of standard substances in relevant drug testing.

## 1. Introduction

Chelerythrine (CH, Figure 1A) is one of the most important members of the quaternary benzophenanthridine alkaloids (QBAs) family. It is derived from medicinal plants, such as *Fagara semiarticulata* [1], *Macleaya cordata* (Willd.) R. Br. [2,3], *Chelidonium majus* L. [4,5,6,7] and *Zanthoxylum nitidum* (Roxb.) DC. [8,9]. As an active ingredient of herbal medicine, CH has a variety of pharmacological effects, such as anti- inflammation [4,9,10,11], anti-microbial [12], anti-virus [13] and anti-tumor [14,15,16,17]. In order to understand the molecular mechanism of pharmacological effects, the interactions between CH and some DNA and protein molecules have been studied by the methods of absorbance and fluorescence spectroscopy, viscometry, calorimetry, cyclic voltammetry and molecular calculation [18,19,20,21,22,23]. These studies demonstrated that environmental factors, especially pH value, have a significant effect on the interaction between CH and DNA or protein [21,22,23]. 

Ethoxychelerythrine (6-ethoxy-5,6-dihydrochelerythrine, ECH, Figure 1b), obtained from the medicinal plants mentioned above [1,2,6,8,9], is one of the dihydrogen derivatives of CH. Since the ethanol recrystallization method is used in the purification process, ECH is regarded as an artifact produced by the reaction between CH and ethanol [2,8]. However, according to the experiments of Lu et al. [24], ECH may exist in natural plants. Currently, both CH and ECH are chemical reference substances for the identification and quality evaluation of Chinese herbal medicines. In *Chinese Pharmacopoeia* [25], ECH is used as a reference substance to identify *Z. nitidum* (Roxb.) DC., and the content of CH in *C. majus* L. crude drugs is stipulated to be no less than 0.02%. 

The structural transformation of CH, ECH and other QBAs in common solvents and their biotransformation in human hepatocytes have been studied by NMR spectroscopy [26] and mass spectroscopy [27], respectively. Miskolczy et al. revealed the thermodynamics of the self-binding of three QBAs including CH and the effect of this process on the absorption and fluorescence behavior [28]. The native fluorescence of CH has been applied in the analysis of medicinal plants by high-performance thin-layer chromatography [5] and microchip electrophoresis [7]. It is reported that the anionic surfactant sodium dodecyl sulfate can change the fluorescence features, resulting in the structural transformation of CH [29]. However, there are rarely reported comparative studies of fluorescence, absorbance and chromatography of CH and ECH, and it remains unclear whether the spectral and chromatographic characteristics of CH and ECH in common protic solvents are the same or different.

Compared with ECH, the molecular structure of CH has a higher degree of conjugation because of the double bond C=N^+^ between N5 and C6. Considering the different structures, their absorbance and fluorescence spectra, CH should have longer wavelength. However, under the same experimental conditions in this study, the fluorescence and absorbance spectra of CH and ECH were found to be identical. This intriguing phenomenon may indicate that these two substances could be transformed into the same structure in the same protic solvent. If so, it will be possible to substitute the standard substances in drug testing. In practice, drug testing is usually carried out in protic solvents (water, methanol, etc.), and the standard substances are usually expensive and sometimes difficult to obtain. If there is a substitute, drug testing will become more economical and easier. Moreover, the chemical changes of biomolecules and the relationship between their molecular structures and spectral characteristics have always been important and interesting topics [30,31,32,33].

The purpose of this work is to illustrate the structural transformation of CH and ECH in methanol and water by combining their spectral changes in these solvents. The fluorescence and absorbance of CH and ECH in methanol and water with various pH values were characterized experimentally, and the fluorescence quantum yield of their products in methanol and alkaline solutions were measured. The result of high-performance liquid chromatography (HPLC) verified that CH and ECH can generate the same product in the same protic solvent.

## 2. Materials and Methods

### 2.1. Materials

Chemical reference substances of CH (serial no. 111718-201402, molecular formula: C_21_H_18_ClNO_4_, molecular weight: 383.82) and ECH (serial no. 110847-200601, molecular formula: C_23_H_23_NO_5_, molecular weight: 393.43) were purchased from the National Institutes for Food and Drug Control (Beijing, China) and dissolved in methanol (chromatographic grade, Tedia, OH, USA) to prepare the stock solutions. The Britton–Robinson buffer solution was a mixture of phosphoric acid, boric acid, and acetic acid (each 0.02 M) and adjusted to the appropriate pH by adding 0.1 M NaOH solution, and all the buffer chemicals were analytically pure. Acetonitrile (chromatographic grade, Duksan, Ansan-si, Korea) and triethylamine (analytically pure, Tianjin Guangfu, Tianjin, China) were used without further treatment. The water used throughout the study was double deionized, which was verified to be non-fluorescent.

### 2.2. Apparatus

Fluorescence measurements were carried out on a fluorescence spectrophotometer (F-7000, Hitachi, Tokyo, Japan) equipped with a xenon lamp and 1 cm quartz cell. The excitation and emission slits (band pass) of 5 nm/5 nm were used, and a 350 nm filter was placed in the emission path to remove secondary spectra. Absorbance spectra were recorded by a spectrophotometer (UV-2501PC, Shimadzu, Tokyo, Japan) with 1 cm quartz cell. The pH values of solutions were measured by a pH meter (868 pH/ISE, Orion, MA, USA). The HPLC tests were conducted on a liquid chromatograph (1260 Infinity II, Agilent, CA, USA) equipped with Agilent 1260 ultraviolet detector. 

### 2.3. General Procedure for Spectral Measurement

An appropriate amount of CH or ECH and buffer solutions were added into a series of 10 mL volumetric flasks, diluted to the mark with methanol or water and mixed thoroughly. The fluorescence or absorbance spectra and pH value of the solutions were measured at room temperature.

### 2.4. Measurement of Fluorescence Quantum Yield

Quantum yield was estimated by a referential method [34,35] with L-tryptophan (quantum yield 0.14) as the reference. L-tryptophan and CH or ECH solutions were prepared at appropriate concentrations to ensure their absorbance (*A*) was no larger than 0.05 in the measurement. After obtaining the absorbance and fluorescence spectra, the quantum yield was calculated according to Equation (1).
(1)$$Y_{u}=Y_{s}\cdot \frac{F_{u}\cdot A_{s}}{F_{s}\cdot A_{u}}$$
where *Y*_u_ and *Y*_s_, *F*_u_ and *F*_s_, and *A*_u_ and *A*_s_ were the fluorescence quantum yield, integral fluorescence intensity and absorbance of the unknown and reference solutions at their excitation wavelengths, respectively.

### 2.5. HPLC Test

In HPLC tests, an Agilent C_18_ chromatographic column (5 μm, 4.6 mm × 150 mm) was used. The solution of acetonitrile-water (*v*/*v* 26:74) containing 1% trimethylamine was used as the mobile phase, and its pH value was adjusted to around 3.0 by phosphoric acid. The flow rate of the mobile phase was 1.0 mL/min, the column temperature was 25 °C, and the detection wavelength was 269 nm.

## 3. Results and Discussion

### 3.1. Three-Dimensional (3D) Fluorescence Spectra of CH and ECH in Methanol Solutions

Figure 1 shows the 3D fluorescence spectra of CH and ECH in methanol solutions. In these spectra, CH and ECH have the same maximum excitation/emission wavelengths (λ_ex_/λ_em_) of 279 nm/400 nm, which indicates that both CH and ECH have the same fluorophore in methanol solutions. Considering the differences in the molecular structures between these two compounds, it is inferred that the existing forms of CH and/or ECH in methanol should have changed.

Considering that the polar C=N^+^ bond of QBAs molecules is characterized by its sensitivity to nucleophilic attack [36], it is easy to form adducts at the C6 atoms when CH is dissolved in protic solvents such as methanol or ethanol (which can ionize protons to produce negatively charged nucleophilic ions). In methanol solution, CH molecules (CH^+^) can react with methanol to form 6-methoxy-5,6-dihydrochelerythrine (MCH), as shown in Figure 2. According to an earlier report by Huang et al. [37], MCH can be obtained from Z. nitidum (Roxb.) DC. by a methanol recrystallization method. Our research proves the existence of MCH, and its formation mechanism may be shown in Figure 2.

On the other hand, CH can be transformed into MCH in methanol solution through a competitive reaction with methoxy and ethoxy groups, as shown in Figure 3. 

Therefore, CH and ECH in methanol solutions could be transformed into MCH by addition reaction or displacement reaction, respectively, resulting in the same 3D fluorescence spectra. The fluorescence quantum yields of CH and ECH in methanol solutions were also estimated to be the same, Y = 0.13, which was in agreement with the above discussion. In further experiments in which the experimental conditions were changed as follows, the CH and ECH methanol solutions still exhibited the same spectral and chromatographic properties, which verified the above conclusion.

### 3.2. 3D Fluorescence Spectra of CH and ECH in Acidic Methanol Solutions

Figure 2 shows the 3D fluorescence spectra of CH and ECH methanol solutions containing 0.01 M H^+^. When excited at either 270 nm or 315 nm, the steady-state fluorescence spectra of these two compounds showed the maximum emission at 550 nm, but their fluorescence peaks at 400 nm presented in Figure 1 almost disappeared. As described in Figure 2, when H^+^ is added to the methanol solution of CH or ECH, the proton ionization of the solvent will be inhibited, resulting in a decrease in the concentration of negatively charged nucleophilic ions, which will promote the reverse reaction of forming CH^+^. Therefore, the fluorescence peak at λ_em_ = 550 nm should belong to CH^+^, and its emission wavelength is longer than that of MCH (λ_em_ = 400 nm), which is attributed to the high conjugation in its molecular structure. Compared with MCH, the fluorescence intensity of CH is much weaker, at less than 1/20 of that of MCH at the same concentration.

### 3.3. Absorbance Spectra of CH and ECH in Methanol–Water Solutions Containing Different Proportions of Water

Figure 3 shows the absorbance spectra of CH and ECH in methanol–water solutions containing different proportions of water. As shown in Figure 3, the CH and ECH solutions have the same absorbance spectra. When the proportion of water increases from zero to 95%, an absorption band at 370–500 nm appears and rises, indicating that MCH (CH and ECH have transformed into MCH) changes into another form with a higher degree of conjugation. Since water can undergo an autoprotolysis reaction to produce H^+^, MCH can be converted into CH^+^ in neutral aqueous solution, as shown in Figure 4.

Figure 4 seems to be the inverse reaction of Figure 2, except that the solvent is different. Since the autoprotolysis constant K_s_ = 10^−14.0^ of water is greater than K_s_ = 10^−16.7^ of methanol, the concentration of H^+^ in water is larger than that in methanol. The structural transformation shown in Figure 4 can move to the right. 

In the molecular structure of MCH, N5 and C6 atoms occupy sp^3^ hybridized orbitals, and are not conjugated with other parts of the molecule, so the whole molecule of MCH is non-planar [38] and it is colorless. When MCH is converted to CH^+^, the N5 and C6 atoms are transformed to sp^2^ hybridized orbitals, which provide p-electron for the conjugation of molecules and display a pale yellow color. 

### 3.4. 3D Fluorescence Spectra of CH and ECH in Aqueous Solutions at Different pH Values

Figure 4 is the 3D fluorescence spectra of CH and ECH in acidic aqueous solutions (pH = 1.9). The fluorescence peaks at λ_em_ = 550 nm in Figure 4 are almost the same as those in Figure 2, but the shape and fluorescence peak intensity in these two figures are slightly different, which is mainly due to the different solvent environments.

Figure 5 is the 3D fluorescence spectra of CH and ECH aqueous solutions with pH 5.7. Compare Figure 5 with Figure 4, the fluorescence peak located at λ_em_ of 550 nm remains almost invariant, but a new fluorescence peak with λ_em_ of about 410 nm has appeared, indicating that a new fluorescent substance has formed.

As shown in Figure 6, with the further increase of the pH value of solutions, the fluorescence peak at λ_em_ = 550 nm in Figure 5 weakens and finally disappears, while the fluorescence peak at λ_em_ = 410 nm is significantly enhanced.

Figure 6 is very similar to Figure 1, but the solvent is different. In alkaline aqueous solutions, CH can react with H_2_O to form 6-hydroxyl-5,6-dihydrochelerythrine (CHOH), as shown in Figure 5.

The reaction of CH^+^ with H_2_O is similar to that of CH^+^ with CH_3_OH described in Figure 2. Both reactions are the electrophilic addition reaction and are sensitive to the pH values of solutions. Increasing the pH value can promote the reaction to proceed in the positive direction, and vice versa. 

The molecular structures of CHOH and MCH are similar, so they have similar fluorescence spectra. A slight difference between them is that λ_em_ of CHOH is about 410 nm, while that of MCH is 400 nm. The fluorescence quantum yields of CH and ECH (actually CHOH) in alkaline solutions at pH = 11.0 were measured, and the same value Y = 0.15 was obtained.

### 3.5. Absorbance Spectra of CH and ECH in Aqueous Solutions at Different pH Values

Figure 7 is the absorbance spectra of CH and ECH aqueous solutions with different pH values. The spectral characteristic of Figure 7 is very similar to that of Figure 3. The appearance of an isosbestic point in the spectra proves that there are only two light-absorbing species, CH^+^ and CHOH, in the solutions, and the interconversion between them occurs with the change of pH value. This spectral characteristic is consistent with the reaction mechanism shown in Figure 5.

### 3.6. Effect of pH Value on Fluorescence Intensity of CH and ECH in Aqueous Solutions

Figure 8 is the excitation and emission spectra of CH and ECH in aqueous solutions at various pH values. The spectra of CH and ECH are very similar, which is consistent with the above discussion. In these spectra, the centers of λ_ex_ and λ_em_ locate at 279 nm and 406 nm, respectively, which is consistent with the 3D fluorescence spectra shown in Figure 6. Changing the pH value will result in the change in fluorescence intensity, but it has no effect on λ_ex_ and λ_em_.

Figure 9 shows the relationship between fluorescence intensity and pH values of the CH and ECH aqueous solutions. These two curves are nearly the same. When pH > 6.0, there is an increase of fluorescence intensity due to the transformation of CH^+^ to CHOH. When pH > 12.0, the decrease in fluorescence intensity is probably due to the proton ionization of the 6-hydroxyl group of CHOH. CHOH does not participate in the conjugate plane of the molecule, and its proton ionization has almost no effect on the fluorescence wavelength and only a slight effect on the fluorescence intensity.

### 3.7. HPLC Chromatograms of CH and ECH in Acidic Aqueous MOBILE phase

To confirm the structural transformation of CH and ECH in protic solvents, CH, ECH and their mixture in methanol were analyzed by HPLC (Figure 10). All three chromatograms show a retention time of about 9.4 min. The same compound has the same retention time under the same chromatographic condition, which is the qualitative basis of HPLC. The above result confirms that CH and ECH transform into the same molecular structure in the experiment, which is consistent with the above conclusion.

## 4. Conclusions

The chemical reference substances CH and ECH have different molecular structures, and this study found that they have the same fluorescence, absorbance and chromatographic properties in protic solvents, because they can react with solvent molecules such as methanol, ethanol and water and transform into the same molecular structure. After CH and ECH were dissolved in methanol, they were converted to MCH by the nucleophilic addition reaction and displacement reaction, respectively. MCH emitted significant fluorescence at the wavelength of 400 nm. When the pH value of methanol solution decreased, MCH was converted into CHz, and the emission wavelength was red-shifted to 550 nm with a fluorescence quenching at 400 nm. When diluted with water, the MCH methanol solution was converted into CH^+^ aqueous solution. As the pH value of the solution increased, CH continued to be converted into CHOH, and strong fluorescence was observed at the wavelength of 410 nm. The CH and ECH methanol solutions had the same chromatographic retention time, as well as the fluorescence and absorption spectra, proving that they were indeed transformed into the same molecular structure. The spectral and chromatographic characteristics and molecular mechanisms of CH and ECH revealed in this work may be applied in the future drug testing. 

## Data Availability

All data generated or analyzed during this study are included in this published article.

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
