# Peer review of "Fluorescence, Absorption, Chromatography and Structural Transformation of Chelerythrine and Ethoxychelerythrine in Protic Solvents: A Comparative Study"

_molecules, 2022, doi:10.3390/molecules27154693_

Round 1

Reviewer 1 Report

In this manuscript Chelerythrine (CH) and Ethoxychelerythrine (ECH) - one of the dihydrogen derivatives of CH – are studied and their optical properties like fluorescence and absorption, and chromatography are reported in different pH environments. These materials are extracted from medicinal plants; and are used as active ingredients of herbal medicine. CH has been considered to have several pharmacological properties. Here a systematic design for the experiment and discussion can be seen, and all the experimental results are elaborated.

The discussion is interesting, well organized, and with important impact of such molecules, it is interesting to have such characterizations; However, to make the discussion more general and useful for a broader range of readers, I’d like to point out a few issues to be considered by the authors:

While the experiments and results are well organized and reported in this manuscript, it is still not clear why they are important. This is vague and the authors should discuss about the necessity of such results in the context of their applications. To this end the authors could also compare their results and methods with similar studies on the optical properties of bio-molecules; what, for example, is done in:

Phys. Chem. Chem. Phys., 2019,21, 26301-26310

J. Phys. Chem. A 2017, 121, 20, 3909–3917

J. Phys. Chem. B 2014, 118, 19, 4983–4992

- I would also suggest to use of unique representation method for fluorescence measurements. The style used in figure 1 could be used in all the cases with a unique color-code; so, one could compare fluorescence signal strength as well.

- A through English proofreading and revision is required.

In the end, given the interest and potential applications, and the quality of this manuscript, I support publication of this manuscript in the journal of Molecules after considering the beforementioned points to reinforce the manuscript.

Reviewer 2 Report

This article mainly introduced the fluorescence features and differences of Chelerythrine (CH) and Ethoxychelerythrine (ECH) in different conditions. However, there are some questions to be resolved before it can be published in this Journal. Detailed comments are as follows:

1.     Two isoflavones formononetin (F) and ononin (FG) are introduced and compared in the abstract. However, there isn’t any word mentioning formononetin, ononin, F or FG in the main body of the article. Instead, Chelerythrine (CH) and Ethoxychelerythrine (ECH) are used throughout the text, which is very confusing.

2.     The authors claimed that CH and ECH can react with protic solvents and both transform into MCH, thus they have same fluorescence in protic solvents. There should be control group fluorescence data of CH and ECH in aprotic solvents to prove that they two are originally different.

3.     In 3.6 Effect of pH in fluorescence intensity of CH and ECH aqueous solutions, is there any special reason why CH solution with double of the concentration of ECH was used?

4.     There is not enough evidence to support the speculated mechanism shown in Scheme 2. To prove nucleophilic addition occurred, NMR data in methanol should be provided.

5.     The chromatograms of HPLC is not evidential enough to make the conclusion like “ the two transformed into the same molecular form during the experiment”.

6.     It is inappropriate to define CH+ as a weak fluorescent substance and define CHOH as  a strong fluorescent substance, since the authors didn’t give any judgment standard about what are strong and weak substances, plus that no quantitative information about their fluorescent information was given.

7.     English should be checked and improved. There are many typo and grammar errors to be corrected. For example, P2L78 “It is conclude that CH and ECH can converted…”; P3L126 “Since the molecular structure of CH and ECH are different…”; P4L142 “ECH in methanol solution can converted to…”; P6L199 “This is because of the solvents are different”; P6L199 “ CH and ECH water solution”; P7L227 “promote the reaction move to …”…

8.     Please check the format of captions, for example Figure 9 and 10.

Round 2

Reviewer 1 Report

The revised manuscript is actually improved; and previous comments are considered. It contains information useful for researchers and potentially for those in industry. Therefore, I would recommend its publication in the journal of Molecules.

Author Response

Our article has been edited for proper English language, grammar, punctuation, spelling, and overall style by one or more of the highly qualified native English speaking editors at 51 runse, which is certified by the following document. The editorial certificate is in the supplementary materials.

Reviewer 2 Report

The authors have well-addressed issues I pointed out in the last version, I have no further questions.

Author Response

Thank you for your recognition. Our article has been edited for proper English language, grammar, punctuation, spelling, and overall style by one or more of the highly qualified native English speaking editors at 51 runse. The editorial certificate is in the supplementary materials.
